# Drug-Induced Immune Thrombocytopenia Toxicity Prediction Based on Machine Learning

**DOI:** 10.3390/pharmaceutics14050943

**Published:** 2022-04-26

**Authors:** Binyou Wang, Xiaoqiu Tan, Jianmin Guo, Ting Xiao, Yan Jiao, Junlin Zhao, Jianming Wu, Yiwei Wang

**Affiliations:** 1School of Pharmacy, Southwest Medical University, Luzhou 646000, China; wangby9028@126.com (B.W.); z18628871125@163.com (J.Z.); 2School of Basic Medical Science, Southwest Medical University, Luzhou 646000, China; tanxiaoqiu@swmu.edu.cn (X.T.); guojm63@163.com (J.G.); xiaoting9853@163.com (T.X.); 3Key Laboratory of Medical Electrophysiology, Southwest Medical University, Luzhou 646000, China; 4Ministry of Education & Medical Electrophysiological Key Laboratory of Sichuan Province, Institute of Cardiovascular Research, Southwest Medical University, Luzhou 646000, China; 5State Key Laboratory of Biotherapy and Cancer Center, West China Hospital, Sichuan University, Chengdu 610041, China; jiaoyan315@163.com; 6Sichuan Key Medical Laboratory of New Drug Discovery and Druggability Evaluation, School of Pharmacy, Southwest Medical University, Luzhou 646000, China; 7Luzhou Key Laboratory of Activity Screening and Druggability Evaluation for Chinese Materia Medica, School of Pharmacy, Southwest Medical University, Luzhou 646000, China

**Keywords:** machine learning, drug-induced immune thrombocytopenia, k-nearest neighbor, structural alert

## Abstract

Drug-induced immune thrombocytopenia (DITP) often occurs in patients receiving many drug treatments simultaneously. However, clinicians usually fail to accurately distinguish which drugs can be plausible culprits. Despite significant advances in laboratory-based DITP testing, in vitro experimental assays have been expensive and, in certain cases, cannot provide a timely diagnosis to patients. To address these shortcomings, this paper proposes an efficient machine learning-based method for DITP toxicity prediction. A small dataset consisting of 225 molecules was constructed. The molecules were represented by six fingerprints, three descriptors, and their combinations. Seven classical machine learning-based models were examined to determine an optimal model. The results show that the RDMD + PubChem-k-NN model provides the best prediction performance among all the models, achieving an area under the curve of 76.9% and overall accuracy of 75.6% on the external validation set. The application domain (AD) analysis demonstrates the prediction reliability of the RDMD + PubChem-k-NN model. Five structural fragments related to the DITP toxicity are identified through information gain (IG) method along with fragment frequency analysis. Overall, as far as known, it is the first machine learning-based classification model for recognizing chemicals with DITP toxicity and can be used as an efficient tool in drug design and clinical therapy.

## 1. Introduction

Drug-induced immune thrombocytopenia (DITP) represents a life-threatening clinical syndrome manifested by a dramatic platelet reduction [1,2,3]. The DITP usually occurs within five to ten days after a patient has been exposed to stimulant drugs, and it is accompanied by severe bleeding and can even lead to death [4]. Many multifarious pathogenic mechanisms have been identified in emblematic DITP cases. Their common feature is severe thrombocytopenia caused by drug-dependent antibodies (DDAbs) that can activate platelet clearance or destroy platelets [3,5]. Thrombocytopenia is a frequent clinical hematologic abnormality in patients brought on by diversified factors [6]. The presence of DDAbs cannot be detected in a patient’s serum or plasma; thus, conventional screening methods are not suitable to determine whether someone is suffering from the DITP. Even when DITP is suspected, physicians cannot accurately determine which medication has caused thrombocytopenia based on clinical information because most patients take multiple medications simultaneously. Currently, in vitro DDAbs tests are considered the most efficient method for DITP detection [2,7]. However, this method has certain disadvantages, including insensitivity to DDAbs antibodies, poor solubility of drugs, difficult detection of drug metabolites that result in DITP, high cost, and tedious data analysis process [4,8]. Meanwhile, a high-precision DDAbs test has stringent requirements for the experimental conditions and equipment and operator expertise. Currently, only a few specialized laboratories in platelet immunology can provide reliable DDAbs test results [2,3,4]. Moreover, DDAbs test cannot assist physicians in providing timely diagnosis and effective treatment to patients.

In clinical practice, a rapid and even life-threatening reduction of platelets can occur as the DITP worsens. In addition, a DITP-causing drug and its metabolites are settling obstinately in a patient’s body, continuing to clear the platelets so that platelet replacement therapy will not have a significant effect on preventing the DITP. Therefore, the most effective therapeutic method against the DITP is the exclusion of all drugs that may cause DITP [9]. The hospitalized patients are at a higher risk for DITP, especially the elderly with metabolic diseases, but it is unfeasible to stop all their medications simultaneously. Thus, an advanced technique that can quickly and accurately determine which drug induces the DITP is crucial for providing patients with precise and convenient treatment.

Compared with experimental methods, in silico methods are less expensive and more efficient; therefore, in silico methods have been widely used in recent years in drug design and development, disease diagnosis, and ADMET (absorption, distribution, metabolism, excretion, and toxicity) prediction [10,11,12,13]. Recently, machine learning-based algorithms have attracted significant attention due to their excellent performance in developing predictive models. For instance, Mansouri et al. developed a number of models to predict the logarithmic acid dissociation constant pKa of compounds using a series of classic machine learning-based methods, including extreme gradient boosting (XGBoost), k-nearest neighbor (k-NN), and support vector machine (SVM), and these models outperformed the corresponding commercial models [14]. Jaganathan and co-workers used the SVM classifier to generate a prediction model of hepatotoxic compounds, and the optimal model outperformed the models reported in the previous studies [15], achieving an overall prediction accuracy of 81.1% and 75.6% on the internal and external validation sets, respectively [16]. In addition, various excellent structure–activity relationship (SAR) models have been developed for quickly determining the properties of compounds by their structures. However, to the best of the authors’ knowledge, there have been no studies on in silico models for the DITP toxicological evaluation of diverse chemicals. Compared with the laboratory approach, the establishment of DITP prediction models using machine learning-based methods could be more convenient and inexpensive. Therefore, it is urgent to develop accurate and reliable in vitro models for DITP toxicity.

In view of this, this study aims to develop a robust predictive model of DITP toxicity using various molecular descriptors and seven classic machine learning-based methods. Internal five cross-validation and external validation were employed to evaluate the predictive capability of the established models. The results show that the k-NN method based on RDKit molecular descriptors (RDMD) and PubChem molecular fingerprint achieves the best performance among all methods. Moreover, the applicability domain (AD) is defined to verify the reliability and reasonability of the best model. Moreover, structural alerts (SAs) of the DITP toxicity are carefully analyzed.

## 2. Materials and Methods

### 2.1. Data Collection and Preparation

The data used in this study were collected from a reliable website called “Platelets on the Web” (https://www.ouhsc.edu/platelets/ (accessed on 20 April 2021)) [17]. This website was built by James N. George and co-workers with the aim to help scientists to understand important substances that could induce thrombocytopenia and thrombotic microangiopathy better. On this website, a DITP dataset has been established by developers, and it includes information on compounds tested on DDAbs by the Blood Center of Wisconsin and those collected from publicly available patient reports [17].

Standard laboratory data are more reliable and accurate than fragmentary clinical cases. Besides that, abundant efforts and time are needed to process original data from different clinicians [18]. Therefore, in this study, only data from the DDAbs tests were included in our dataset. The raw DDAbs data included information on agents, compound preparation, and food. To ensure high-quality data for subsequent analysis, data cleaning was conducted. The raw data were preprocessed through the following steps. First, data containing mixtures and duplicate drugs were removed. Next, data including polysaccharides and peptide-based macromolecules were removed. Finally, data on food were removed. After the data collection, sorting, and cleaning processes, the final DITP dataset, containing information on 225 diverse drugs, was established for further investigation. In this dataset, drugs with detectable DDAbs antibodies were defined as DITP toxicants, and the remaining drugs were defined as DITP non-toxicants. Thus, the final dataset included 93 DITP toxicants and 132 DITP non-toxicants. This dataset was randomly divided into a training set containing 180 compounds and an external validation set containing 45 compounds according to a ratio of 8:2, as shown in Table 1. The detailed information on each drug in the dataset included its chemical name and Simplified Molecular Input Line Entry System (SMILES), as given in Appendix A.

### 2.2. Molecular Feature Calculation

To describe molecule structures, six types of molecular fingerprints were employed: MDL Molecular Access fingerprint (MACCS, 166 bits), PubChem fingerprint (PubChem, 881 bits), CDK fingerprint (CDK, 1024 bits), CDK extended fingerprint (ExtFP, 1024 bits), Klekota-Roth fingerprint (KRFP, 4860 bits), and Atom Pairs 2D fingerprint (AP2D, 780 bits). The sizes and pattern types of fingerprints are listed in Appendix A. All fingerprints were calculated using PaDEL-Descriptor software (version 2.21, Chunwei Yap, Singapore) [19]. In addition, to characterize the molecules more accurately, three datasets of molecular descriptors (MD), including 13MD, RDMD and Chemical Checker (CCMD), were calculated and used to represent the physicochemical and biological properties of chemicals. The 13MD contained 13 types of commonly used molecular descriptors: molecular solubility, molecular weight, octanol-water partitioning coefficient, apparent partition coefficient at pH = 7.4, number of hydrogen bond donors, sum of the oxygen and nitrogen atoms, number of hydrogen bond acceptors, number of rotatable bonds, number of rings, number of aromatic rings, molecular surface area, polar surface area and molecular fractional polar surface area. Thirteen descriptors were chosen because they have been widely used in the prediction of compound properties and have provided excellent prediction performances [20,21,22]. The RDMD contained 200 molecular descriptors, including physicochemical properties and structure characteristics, which have been used in many studies and have achieved satisfactory results [23,24,25,26,27]. The CCMD is a novel type of biological descriptor containing 25 various bioactive spaces [28]. Moreover, the simple representation of CCMD is compatible with different types of computational tools in a multi-dimensional form. Compared with other chemical descriptors, the CCMD has the unique advantage of bioactivity [28]. The three datasets of molecular descriptors were calculated by Discovery Studio 3.1, the *Descriptors* module of the Python RDKit package version 2017.09 (https://github.com/rdkit/rdkit (accessed on 13 May 2021)), and the *Signaturizer* package version 1.1.10 in Python (http://gitlabsbnb.irbbarcelona.org/packages/signaturizer (accessed on 16 July 2021)), respectively.

Pervious research demonstrated the importance of dimensionality reduction for toxicity prediction, which can enhance prediction performance, increase interpretability, and reduce computational complexity [16,29]. Thus, eliminating redundant and irrelevant features is considered desirable for our prediction modeling. To simplify the molecular descriptor data, the null value, extreme numerical values, and descriptors with eigenvalues or variance of zero were removed. Then, the correlation between descriptors was analyzed, and duplicate descriptors with a pairwise correlation greater than 0.95 were eliminated. After streamlining the descriptors, 10 out of 13 descriptors remained in the 13MD set, and the newly obtained dataset was denoted by 10MD. For the RDMD and CCMD, 141 and 128 descriptors were left for modeling, respectively. Finally, since values of different descriptors could have different ranges of values, their values were normalized to the range (0, 1) by:(1)$$x^{*}=\frac{x-\min}{\max-\min}$$
where *x* is the original value, *x*^∗^ is the normalized value, and *max* and *min* are the maximum and minimum values of a descriptor, respectively.

### 2.3. Machine Learning-Based Methods

In this study, seven machine learning-based methods, including the SVM [30], k-NN [31], random forest (RF) [32], naive bayes (NB) [33], artificial neural network (ANN) [34], adaptive boosting (AdaBoost) [35], and XGBoost [36], were used to build binary classification models. The detailed descriptions of these methods can be found in the corresponding literature. To select an optimal combination of molecular fingerprints and descriptors, various combinations were used to develop basic models based on each of the seven algorithms. To achieve the best performances of the constructed models, their hyperparameters were optimized by the five-fold cross-validation method on the training set. Moreover, to reduce the variance caused by random partitioning, cross-validation was repeated 10 times for each model. For the SVM, the radial basis function (RBF) was used as a kernel, and the penalty parameter *C* was set to different values. For the k-NN, the nearness was measured by the Euclidean distance-based metrics, and various numbers of neighbors were considered. For the RF, the maximum depth (max_depth) was set to four, and the best split was calculated using the number of trees in the forest and the number of features. For the ANN, the parameter solver was set to stochastic gradient descent (sgd), and the hidden layer size was adjusted. For the AdaBoost, the number of estimators was optimized. For the XGBoost, the maximum depth of a tree and the minimum sum of the instance weight needed in a child were optimized. We used the default values in *scikit-learn* for NB and other parameters not mentioned. The hyperparameters’ values are given in Appendix A. The SVM, k-NN, RF, NB, ANN, and AdaBoost models were implemented in the *scikit-learn* package of Python (version 0.23.2) [37]. The XGBoost model was built by the *XGBoost* package version 1.4.2 in Python (https://github.com/dmlc/xgboost (accessed on 28 June 2021)).

To address the bias problem in the training set, the synthetic minority oversampling technique (SMOTE) [38], a data augmentation approach that has been widely used in previous research, was applied in this study [39,40,41,42]. The SMOTE was accomplished by the *imbalanced-learn* package [43] (vision 0.8.0).

### 2.4. Evaluation Metrics

The prediction performances of all models were assessed by four evaluation indicators: sensitivity (*SE*), specificity (*SP*), accuracy (*ACC*), and Matthews correlation coefficient (*MCC*). SE and SP denote the prediction accuracies of DITP toxicants and DITP non-toxicants, respectively; *ACC* indicates the proportion of correctly predicted DITP toxicants and DITP non-toxicants among all detected compounds; *MCC* is generally regarded as a balanced measurement indicator, and it is not affected by positive and negative disproportionality in a dataset. The *MCC* value is between −1 and 1, where 1 indicates perfect prediction, zero indicates the level of random prediction, and −1 indicates complete disagreement in prediction. The evaluation indicators of *ACC*, *SE*, *SP*, and *MCC* are respectively calculated by:(2)$$ACC=\frac{TP+TN}{TP+FP+TN+FN}$$
(3)$$SE=\frac{TP}{TP+FN}$$
(4)$$SP=\frac{TN}{TN+FP}$$
(5)$$MCC=\frac{TP\times TN-FP\times FN}{\sqrt{\left(FP+TN\right)\left(FP+TP\right)\left(FN+TN\right)\left(FN+TP\right)}}$$
where *TP* denotes the number of true positives, *TN* denotes the number of true negatives, *FP* is the number of false positives, and *FN* is the number of false negatives.

The AUC represents the area under the receiver operating characteristic curve, which is a common indicator of a classification model. The AUC allows a comprehensive assessment of the model’s ability to classify negative and positive compounds, even in the presence of a positive and negative data imbalance [44]. Therefore, the AUC has been widely used for evaluating the quality of various machine learning-based models. In this study, the AUC denoted an important assessment metric of the models’ prediction performances.

### 2.5. Applicability Domain Definition

According to the Organization for Economic Co-operation and Development (OECD) principles for SAR models, a SAR model should define its AD strictly [45,46]. AD analysis can help to understand whether a developed SAR model is suitable for specific sets of data. In this study, a method based on the Euclidean distance was applied to identify the AD of the developed models. This method has been commonly used and has been considered the most useful distance measure in SAR-related studies [47]. This method compares the Euclidean distances between compounds and a dataset with a predefined distance threshold. If compounds’ Euclidean distance values are larger, it is considered that these compounds are outside the AD of the model and have lower prediction accuracy than compounds with Euclidean distance values smaller. This work was conducted by the AMBIT Discovery software (version 0.04) (http://ambit.sourceforge.net (accessed on 25 August 2021)), and the threshold was set to 95% in the training set to determine the domain of the model [48].

### 2.6. Structural Alerts Analysis

To understand an important structural fragment related to the DITP toxicants better, SAs responsible for this toxic effect were identified by the *IG* method and substructure fragment frequency analysis. The fragments of all compounds were derived from the KRFP [49]. The *IG* value represents an indicator for evaluating the importance of a structural fragment for a classification system. A high *IG* value of a fragment indicates that the fragment is critical for the classification system [50]. The *IG* value was calculated by:(6)$$IG=\mathrm{Ent}\left(D\right)-\sum_{V=0,1}^{V}\frac{D^{V}}{D}Ent\left(D^{V}\right)$$
(7)$$\mathrm{Ent}\left(D\right)=-\sum_{K=0,1}^{K}P_{k}\log_{2}P_{k}$$
where *V* is a value of a fragment, which can be zero or one; *D* represents the number of compounds; *D^V^* represents the number of compounds with or without the fragment; *Ent*(*D*) and *Ent*(*D^V^*) represent the information entropy and conditional entropy of compounds, respectively; *K* represents the classes of compounds as zero or one (zero represents non-toxicant and 1 represents toxicant); and *P_k_* denotes the ratio of each class of compounds.

If a fragment appeared more frequently in DITP toxicants than in DITP non-toxicants, it was considered an SA of DITP toxicants. The frequency of occurrence of a fragment was calculated by:(8)$$\mathrm{Frequency}\;\mathrm{of}\;a\;\mathrm{fragment}=\frac{N_{\mathrm{fragment}\_P}\times N_{\mathrm{total}}}{N_{\mathrm{fragment}\_\mathrm{total}}\times N_{P}}$$
where *N_fragment_P_* is the number of compounds containing the fragment in toxicants; *N_total_* is the total number of compounds; *N_fragment_total_* is the total number of compounds containing the fragment; and *N_P_* is the number of toxicants.

## 3. Results and Discussion

### 3.1. Dataset Analysis

To develop a reliable model, the quality of the dataset was analyzed. The chemical space distribution was investigated by calculating the MW and Ghose-Crippen LogKow (AlogP) of the training and external validation sets. The scatter diagrams of the training and external validation sets are presented in Figure 1. As shown in Figure 1, for both sets, the MW values were mainly below 550, and the AlogP values were mainly between −5 and 5. This demonstrated that the two sets shared a similar chemical space and had a good distribution consistency, which is important for establishing a stable prediction model. Moreover, physicochemical properties can provide a great deal of information to help distinguish between toxicants and non-toxicants. In this study, the distribution of six key physicochemical properties of DITP toxicants and DITP non-toxicants, including MW, AlogP, nHBA, nHBD, nRot, and nRing, was investigated. As shown in Figure 2, the histogram distributions of the molecular properties of toxicants and non-toxicants overlapped. Since the difference in each physicochemical property between toxicants and non-toxicants was not significant (*p*-values greater than 0.05), it was impossible to distinguish DITP toxicants from DITP non-toxicants based on simple molecular properties alone.

To explore the chemical diversity of the data further, the Tanimoto similarity index was calculated using the MACCS fingerprint. The Tanimoto similarity index distributions of the training and external validation sets are shown in Figure 3. The average Tanimoto similarity index values of the training and external validation sets were 0.347 and 0.378, respectively, which illustrates the structural diversity of the dataset.

### 3.2. Development of DITP Toxicant Prediction Models

To find an optimal model for predicting DITP toxicants, seven classical and commonly used machine learning-based methods, the SVM, k-NN, RF, NB, ANN, AdaBoost, and XGBoost, were used to construct a series of models and make a direct comparison. Six types of molecular fingerprints and three sets of molecular descriptors were constructed to characterize agents in the dataset. To compare the predictive power of all models fairly, a five-fold cross-validation of each model was carried out on the training set. In total, more than 828 machine learning-based models (including hyper parameter tuning) were evaluated to obtain the best prediction model. The prediction performance of the optimal classification model of each machine learning-based method on the training set obtained by the five-fold cross-validation is given in Appendix A. As shown in Appendix A, different models had different robustness and stability. Overall, the average AUC of all prediction models ranged from 0.509 to 0.628. The average values for ACC, SE, SP, and MCC ranged from 49.9% to 62.7%, 43.4% to 69.0%, 42.1% to 76.3%, and 0.018 to 0.261, respectively, for all models. As shown in Appendix A, there were significant differences in the prediction performance between the models. Therefore, it is necessary to combine different machine learning methods with various characterization approaches of chemical structure to develop comprehensive prediction models for DITP toxicants.

Among all constructed models, the five models (RDMD + PubChem-k-NN, 10MD + PubChem-k-NN, CCMD + KPFP-XGBoost, CCMD + MACCS-XGBoost, and CCMD + PubChem-XGBoost) that yielded the highest AUC values were selected for further analysis, and their prediction performances are listed in Table 2. For these five models, the AUC values were in the range of 0.612–0.628, the ACC values were in the range of 60.7–62.7%, the MCC values were in the range of 0.226–0.261, the SP values were in the range of 56.6%–61.5%, and the SE values were in the range of 61.0–69.0%. For the machine learning-based methods, the k-NN and XGBoost outperformed the other methods in predicting DITP toxicants under the same conditions. Three out of the top-five models were described by the CCMD molecular descriptor with different molecular fingerprints, indicating that the CCMD could accurately represent the property of DITP toxicants. This is logical because the CCMD contains effective bioactivities of the compounds, which can characterize the structure–activity relationship in more detail than other commonly used descriptors. The three models based on the PubChem molecular fingerprint combined with three different types of molecular descriptors suggested that the PubChem was an appropriate selection for characterizing the structure of DITP toxicants. The excellent performance was related to the key information on molecular structures included in the PubChem molecular fingerprint. The PubChem fingerprints had a length of 881 bytes, which encompassed a variety of different molecular structures and molecular features, such as element counts, ring types and counts, atomic pairs, and atomic environments [51,52]. The results showed that the models based on the CCMD and PubChem descriptors could be more effective than other seven descriptors in discovering the relationship between chemical structure and DITP toxicity. Moreover, the model developed based on the k-NN method and described by the combination of the PubChem molecular fingerprint and the RDMD molecular descriptor achieved the best prediction performance among all constructed models. This model had an average prediction ACC of 62.7%, an average MCC of 0.261, an average AUC of 0.628, an average SP of 56.6%, and an average SE of 69.0%, which demonstrated that the RDMD + PubChem-k-NN model had better predictive ability for distinguishing DITP toxicants and DITP non-toxicants than other models. Although the CCMD molecular descriptor integrated with other molecular fingerprints provided a comparatively good in describing the DITP toxicants, the k-NN model based on a combination of the RDMD and PubChem achieved the best prediction result, indicating that not only the machine learning algorithms or molecular fingerprints and descriptors but also their combinations have a considerable influence on the development of the optimal model for predicting DITP toxicants.

Furthermore, to determine whether the models based on various combinations of molecular representations and machine learning-based methods are advantageous, the molecular fingerprints and molecular descriptors in the five top models were used separately to generate models. The detailed results of the optimal models on the training set obtained by the five-fold cross-validation are presented in Table 3. As shown in Table 3, for the k-NN models, the model characterized by the PubChem molecular fingerprint showed better performance than the other two k-NN models based on only one molecular feature. The k-NN-PubChem model had an average SE of 66.9%, an average ACC of 61.2%, an average MCC of 0.231, and an average AUC of 0.614. For the other two k-NN models, the k-NN-10MD and k-NN-RDMD models, the values of the corresponding indicators were relatively lower, especially those of the k-NN-RDMD. Compared to k-NN models using only one descriptor, the k-NN model using the combination of the PubChem molecular fingerprint and RDMD descriptors had an exceptional performance regarding the four predictive indicators of SE, MCC, ACC, and AUC. For the XGBoost models, the combination of CCMD descriptors and KPFP molecular fingerprint significantly improved the prediction performance of the models established by only CCMD descriptors or the KPFP fingerprint. The CCMD + KPFP-XGBoost model had an average SE of 61.9%, an average SP of 61.4%, an average MCC of 0.233, an average ACC of 61.1%, and an average AUC of 0.617, and all five indicators outperformed CCMD-XGBoost and KPFP-XGBoost models. The above-mentioned results clearly demonstrated the enhanced prediction capability of the models using a combination of optimal molecular fingerprints and descriptors. The prediction performance of the k-NN model using the molecular features PubChem + RDMD was relatively better than those using other combinations of molecular fingerprints and descriptors. Consequently, it can be concluded the k-NN algorithm and a combination of the RDMD molecular descriptor and PubChem molecular fingerprint are suitable for DITP toxicity modeling. The k-NN algorithm is a non-parametric method for measuring the distance between different feature values for classification. Because of its simplicity, the k-NN method has been frequently used for building classification models. In addition, the RDMD molecular descriptor can provide valuable information on physicochemical properties and structural features related to the DITP toxicants. The RDMD molecular descriptor with a simple form (141D-vectors) can be easily combined with the features of molecular structures characterized by the PubChem molecular fingerprint. Thus, by combining the k-NN method with the molecular features of RDMD and PubChem, the established model can achieve excellent performance in DITP toxicity prediction.

### 3.3. Verification of Prediction Models on External Validation Set

Next, the external validation set containing 45 molecules was used to explore the generalization capability of the top-five models, which achieved the best performances in predicting DITP toxicants among all the models according to the five-fold cross-validation results. The prediction performances of the five models on the external validation set are shown in Table 4, where the RDMD + PubChem-k-NN model had superior prediction performance on the external validation set over the other four models. The RDMD + PubChem-k-NN model had an AUC of 0.769, SE of 83.3%, SP of 70.4%, ACC of 75.6%, and MCC of 0.526. Compared to the other four models, this model achieved the highest average SE and AUC values, indicating that the RDMD + PubChem-k-NN model could distinguish DITP toxicants from DITP non-toxicants at a reasonable accuracy. The CCMD + KPFP-XGBoost model had an optimal ACC value of 77.8% due to its relatively high SP value. However, the SE value of the CCMD + KPFP-XGBoost model was significantly lower than that of the RDMD + PubChem-k-NN model. The RDMD + PubChem-k-NN model could accurately identify 15 out of 18 DITP toxicants. The SE has been considered as the “gold standard” for assessing model capability to predict positive compounds. The results demonstrated that the RDMD + PubChem-k-NN model had excellent performance in identifying the DITP toxicants on data outside of the training set, indicating that this model had strong robustness. In addition, the RDMD + PubChem-k-NN model also performed best in the five-fold cross-validation. Thus, the RDMD + PubChem-k-NN model could accurately distinguish DITP toxicants from non-toxicants not only on the training set but also on the external validation set.

To explore the reliability of the RDMD + PubChem-k-NN model prediction results, this model’ AD was defined. The Euclidean distance method was applied to evaluate the AD. The statistical results are given in Table 5, where only four compounds were outside of the AD. The AD coverages for the training and the external validation sets were 100% and 91.1%, respectively. The results demonstrated that the prediction results of the RDMD + PubChem-k-NN model on the external validation set were reliable.

Since the consensus prediction is made based on multiple different models, it may be capable of capturing the relationship between the chemical structures of the molecules and the endpoint more efficiently than a single model. To explore whether the consensus model could improve the predictive performance of the single model in this work, a consensus model was developed by simply averaging the predictions for the external validation set given by the top-five models shown in Table 4. Considering the AUC for the external validation set, the consensus model (AUC = 0.731) outperformed only one individual model in the top-five models. Compared with RDMD + PubChem-k-NN model, consensus model showed relatively poorer performance, especially the SE value.

As discussed above, the RDMD + PubChem-k-NN achieved the best and most reliable predictive performance for the external validation set. To investigate why the RDMD + PubChem-k-NN model could identify DITP toxicants more accurately from their chemical structure than the other models further, the predictive results of the RDMD + PubChem-k-NN model and CCMD + KPFP-XGBoost model, which was ranked as the second-best performing model, were compared on the external validation set. The comparison results are shown in Appendix A. Eleven DITP toxicants were correctly identified by both models. Among them, four toxicants were Beta-lactam antibiotics containing β-lactam ring, as shown in Figure 4a, and the other toxicants contained acylamide, aminocarbonyl, quinoline ring, or steroidal, in addition to other substructures. However, four compounds were misclassified by the CCMD + KPFP-XGBoost model, but they were accurately predicted by the RDMD + PubChem-k-NN model. As shown in Figure 4b, these four compounds contained alkyl sulfonyl, piperidinol, benzimidazole, and other substructures. Since it is expected that the number of correlation structures in the training set affects the model’s ability to learn the corresponding features well, the training set was searched for compounds that contained the relevant substructures. It was discovered that a total of 16 compounds in the training set contained the β-lactam ring, suggesting that adequate information on highly-represented substructures was available to both models to achieve good prediction performances in identifying DITP toxicants with these substructures. Still, only three compounds in the training set contained benzimidazole. However, even in the case of insufficient information about these compounds in the training set, the RDMD + PubChem-k-NN model could extract more comprehensive information from these compounds than the CCMD + KPFP-XGBoost model. The above analysis demonstrates that the RDMD + PubChem-k-NN model is the best classifier of DITP toxicants and DITP non-toxicants among all constructed models. As a first attempt, we developed a robust machine learning-based classification model for predicting DITP toxicants from qualitative research. Once the descriptors possessing high correlation with the performance of our model are discovered by analyzing Pearson Correlation Coefficient between DITP toxicity and chemical descriptors. The quantitative structure–activity relationship (QSAR) model for DITP toxicants could also be constructed. It is reasonable to consider that our model has a potential of compatibility with classical QSAR model.

### 3.4. Misclassified Compounds Analysis

Among the constructed models, the RDMD + PubChem-k-NN model had the best prediction performance on the external validation set, but there were still eleven misclassified compounds in the external validation set. The detailed structures of the misclassified compounds are displayed in Appendix A. There could be two reasons for this situation. First, the constructed small-sized dataset affected the prediction capability of models. Second, under a limited number of descriptors, it could be difficult to characterize all agents accurately.

Next, the structures of misclassified compounds were analyzed, as shown in Figure 5. In addition, the analysis results showed that some of the scaffolds existed in both DITP toxicants and DITP non-toxicants. For instance, two DITP non-toxicants (Budesonide and Prednisone), which were misclassified as DITP toxicants on the external validation set, and two DITP toxicants (Methylprednisolone and Dexamethasone) shared a similar scaffold. Thus, some structural features of DITP toxicants are not prominent, which can result in DITP toxicant misclassification. In addition, most of the misclassified compounds contained specific stereoisomeric that could significantly affect the activity of agents. However, in this study, there were no molecular fingerprints and descriptors to characterize the structure of stereoisomers. Tautomerism and protonation of the studied molecules should also be taken into account, as different tautomeric form and protonation state of the same molecule might have completely opposite DITP toxicity. Accurate prediction results of tautomeric form and protonation state depend on the precise description of the structure of a compound. In order to investigate the effects of the fingerprints and descriptors used in our model on the same molecule with different tautomeric form and protonation state, its corresponding fingerprints and descriptors compositions of a typical misclassified compound (Levetiracetam) were analyzed. The comparison results are shown in Appendix A, unraveling different chemical fingerprints and descriptors between the misclassified compound and its tautomeric form and protonation state. Thus, the selected features could characterize the structure of tautomerism and protonation. However, due to the lack of DITP toxicity labels for the tautomeric forms and protonation states of all the drugs in our dataset, it is currently impossible to predict DITP toxicity of a compound with different tautomeric form and protonation state. In the future, these issues will be addressed by enriching drug data, developing more precise descriptors, and applying other in silico methods.

### 3.5. Identification of SAs to DITP Toxicity

To investigate the privileged fragments associated with the DITP toxicity, the IG and frequency analysis substructure of the KRFP were performed to identify SAs. Only the fragments appearing more than six times in the dataset were analyzed [53,54]. The distribution of IG values for each fragment is shown in Figure 6, where the IG values of all 4860 fragments were from zero to 0.029, and the IG values of most fragments were under 0.001. By analyzing the IG values and frequencies of fragments, five SAs and representative toxic compounds were selected. As shown in Table 6, the substructures with high IG values appeared more frequently in DITP toxicants than in DITP non-toxicants. Therefore, the five structural fragments could be considered as SAs to DITP toxicity. Although frequency analysis has been widely used, it cannot describe the spatial arrangement of identified fragments, and it can be difficult to determine toxicants when two or more structural alerts exist in a compound simultaneously. Despite the shortcomings of this method, fragments identified in this study could provide visual alerts useful for DITP toxicity prediction.

## 4. Conclusions

In this study, a reliable dataset containing 93 DITP toxicants and 132 DITP non-toxicants was constructed. Using this dataset and seven machine learning-based methods, including SVM, k-NN, RF, NB, ANN, AdaBoost, and XGBoost, several binary classification models of DITP toxicity were constructed. Six types of molecular fingerprints and three sets of molecular descriptors are used for the characterization of chemicals. A series of cross-validation and external validation tests confirmed the high effectiveness and outstanding performance of the k-NN models in DITP toxicant identification. The RDMD + PubChem-k-NN model has the best prediction performance among all models, achieving an AUC value of 0.769, ACC value of 75.6%, SE value of 83.3%, SP value of 70.4%, and MCC value of 0.526 on the external validation set. Further analysis of the AD definition demonstrates that the prediction ability of the RDMD + PubChem-k-NN model is reliable. The RDMD + PubChem-k-NN classification model has the strongest robustness among all established models and thus can be employed as an alternative method for the qualitative prediction of DITP toxicity. Even so, we have to acknowledge that our model of DITP toxicity is still plenty of room for further improvement. For one thing our model is unable to generalize for each toxicology screening, since high performance of machine learning method depends on several inherent factors, including high quality dataset, appropriate characterization, and rigorous algorithm; for another further experimental and clinical studies are required to confirm our approach. How to overcome or reduce the limitations remains an area for further studies. With the rapid development of novel algorithms and experimental techniques, more accurate and explainable DITP toxicity prediction models will be established soon.

## Figures and Tables

**Figure 1 pharmaceutics-14-00943-f001:**
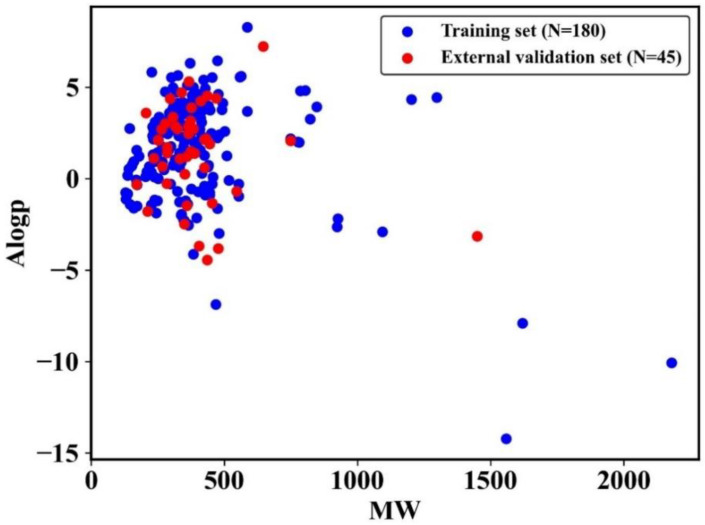
Chemical space distribution of compounds in the training and external validation sets.

**Figure 2 pharmaceutics-14-00943-f002:**
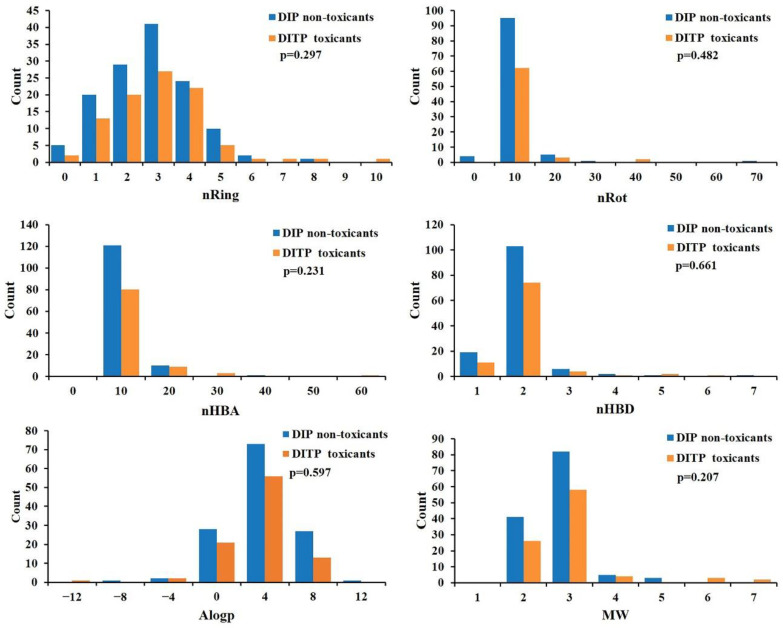
Distributions of six molecular properties of DITP toxicants and DITP non-toxicants.

**Figure 3 pharmaceutics-14-00943-f003:**
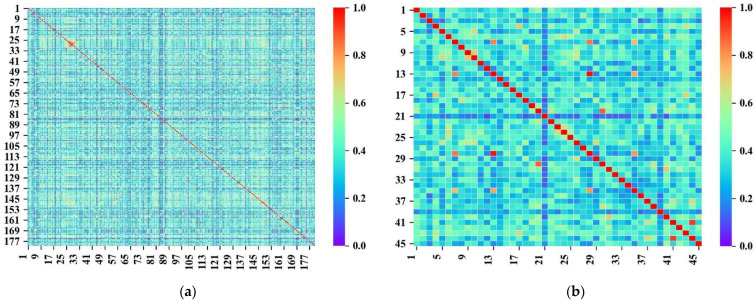
(**a**) Heat map of the Tanimoto similarity index on the training set. (**b**) Heat map of the Tanimoto similarity index on the external validation set.

**Figure 4 pharmaceutics-14-00943-f004:**
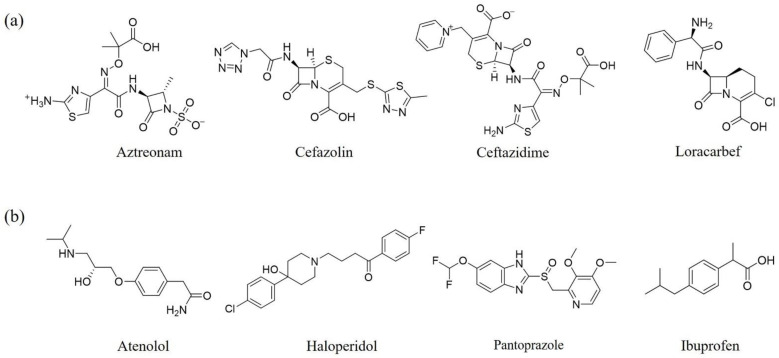
(**a**) The structure of the DITP toxicants with the β-lactam ring correctly identified by the CCMD + KPFP-XGBoost and RDMD + PubChem-k-NN models. (**b**) Structure of DITP toxicants misclassified by the CCMD + KPFP-XGBoost model.

**Figure 5 pharmaceutics-14-00943-f005:**
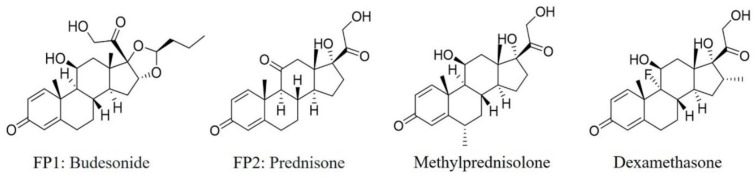
Structure of two false positives and two DITP toxicants.

**Figure 6 pharmaceutics-14-00943-f006:**
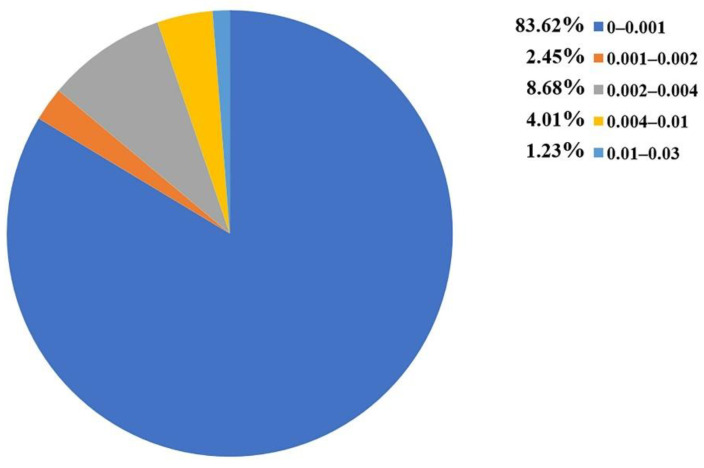
The information gain value distributions of the KRFP fragments.

**Table 1 pharmaceutics-14-00943-t001:** Number of compounds in the training and external validation sets.

	Training Set	External Validation Set	Sum
Toxicants	75	18	93
Non-toxicants	105	27	132
Total	180	45	225

**Table 2 pharmaceutics-14-00943-t002:** The five-fold cross-validation results of the top-five classification models.

Model	Molecular Features	SE (%)	SP (%)	ACC (%)	MCC	AUC
k-NN	RDMD + PubChem	69.0 ± 2.3	56.6 ± 2.1	62.7 ± 1.0	0.261 ± 0.022	0.628 ± 0.011
k-NN	10MD + PubChem	66.9 ± 2.0	57.2 ± 2.4	61.8 ± 1.7	0.243 ± 0.031	0.621 ± 0.016
XGBoost	CCMD + KPFP	61.9 ± 2.7	61.4 ± 3.3	61.1 ± 1.2	0.233 ± 0.025	0.617 ± 0.013
XGBoost	CCMD + MACCS	64.2 ± 2.7	58.3 ± 2.1	60.7 ± 0.8	0.226 ± 0.023	0.613 ± 0.011
XGBoost	CCMD + PubChem	61.0 ± 1.1	61.5 ± 2.1	60.8 ± 0.9	0.226 ± 0.021	0.612 ± 0.010

**Table 3 pharmaceutics-14-00943-t003:** The five-fold cross-validation results of the optimal classification models based on only molecular fingerprints or descriptors.

Model	SE (%)	SP (%)	ACC (%)	MCC	AUC
k-NN-10MD	66.1 ± 1.6	56.2 ± 3.0	61.0 ± 1.9	0.224 ± 0.035	0.612 ± 0.018
k-NN-RDMD	47.6 ± 2.6	67.0 ±2.8	56.9 ± 1.5	0.150 ± 0.034	0.573 ± 0.016
k-NN-PubChem	66.9 ± 2.5	56.0 ± 2.0	61.2 ±1.3	0.231 ± 0.027	0.614 ± 0.014
XGBoost-CCMD	60.5 ± 3.2	60.1 ± 2.8	60.2 ± 2.0	0.207 ± 0.039	0.603 ± 0.019
XGBoost-PubChem	57.5 ± 3.7	62.1 ± 5.5	59.4 ± 1.7	0.196 ± 0.031	0.598 ±0.016
XGBoost-KPFP	57.1 ± 2.5	60.8 ± 3.0	58.1 ± 2.2	0.180 ± 0.046	0.589 ± 0.023
XGBoost-MACCS	53.9 ± 2.5	62.1 ± 2.7	57.8 ± 1.2	0.162 ± 0.031	0.580 ± 0.015

**Table 4 pharmaceutics-14-00943-t004:** Performances of the top-five classification models and consensus model on the external validation set.

Model	Molecular Features	SE (%)	SP (%)	ACC (%)	MCC	AUC
k-NN	RDMD + PubChem	83.3	70.4	75.6	0.526	0.769
XGBoost	CCMD + KPFP	66.7	85.2	77.8	0.531	0.759
k-NN	10MD + PubChem	83.3	63.0	71.1	0.456	0.731
XGBoost	CCMD + PubChem	61.1	85.2	75.6	0.481	0.731
XGBoost	CCMD + MACCS	61.1	81.5	73.3	0.436	0.713
Consensus model	/	61.1	85.2	75.6	0.481	0.731

**Table 5 pharmaceutics-14-00943-t005:** Number of drugs inside and outside of the AD.

	Inside		Outside		AD Coverage (%)
	P	N	P	N	
Training set	75	105	0	0	100
External validation set	18	27	4	0	91.1

**Table 6 pharmaceutics-14-00943-t006:** Five structural alerts of DITP toxicity and their representative structures.

Structure	IG	Freq _P	Freq _N	Representative Structure
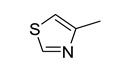	0.0170	2.12	0.21	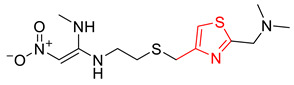
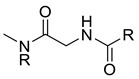	0.0259	1.84	0.41	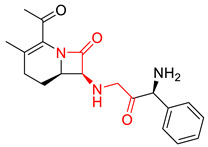
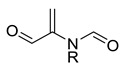	0.0123	1.73	0.49	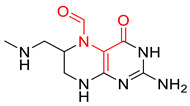
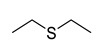	0.0138	1.66	0.54	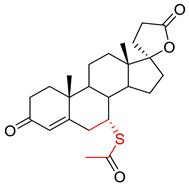
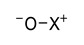	0.0073	1.61	0.57	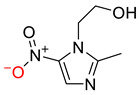

X^+^ represents nitrogen positive atoms, sulfur positive ions, and metal ions.

## Data Availability

The data presented in this study are available in the Appendix A of the paper.

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
