# Peer review of "Drug-Induced Immune Thrombocytopenia Toxicity Prediction Based on Machine Learning"

_pharmaceutics, 2022, doi:10.3390/pharmaceutics14050943_

Round 1
Reviewer 1 Report
The manuscript “Drug-induced immune thrombocytopenia toxicity prediction based on machine learning” is devoted to a relevant and important problem and can be published in “Pharmaceutics” after taking into account the following comments:
- In the introduction, where the authors describe the prospects of in silico methods (lines 75–95), it makes no sense to refer to works [14–22] that are not directly related to this article, it is enough to refer to 1-2 reviews devoted to the QSAR methodology.
- The meaning of references [35 – 39] in line 169 is not clear.
- It is surprising that the authors did not try to use consensus models built on the basis of the top 5 models (Table 4). This could significantly improve the accuracy of DITP predictions of toxicants.
- In the last row of Table 6, the authors erroneously indicate the fragment R-O-, where R is a hydrocarbon radical. It is more correct to indicate N+ - O- (a fragment of the nitro group).
Reviewer 2 Report
Article entitled “Drug-induced immune thrombocytopenia toxicity prediction based on machine learning” is about a machine learning-based classification model for recognizing chemicals with DITP toxicity and can be used as an efficient tool in drug design and clinical therapy. Wang et al., developed a robust predictive model of DITP toxicity using various molecular descriptors and seven classic machine learning-based methods. This article is interesting and may be considered for publication
Comments
- Line no 115: The main concern is that the database is old “(https://www.ouhsc.edu/platelets/) the last update is on 5.4.2015,
- Also, no update on DITP data https://ouhsc.edu/platelets/ditp.html after 2018. Suggest adding more data in future
- No of DITP toxicants is less 93
Reviewer 3 Report
In the manuscript, Wang and co-workers present a machine learning-based method for predicting the toxicity of drug-induced immune thrombocytopenia (DITP). The authors start with a small dataset of molecules, which they encode as fingerprints, descriptors, and their combinations. They then use multiple machine learning-based models to derive models and finally select a final model with the best predictive performance.
The submitted manuscript reports a very interesting approach and could be published after incorporating some minor changes.
1. Please discuss in more detail how your approach is compatible with classical linear or non-linear QSAR approaches and where you see synergies between them.
2. Your approach uses molecular structures as an input for computing the fingerprints and descriptors. Please comment on the extent to which minor errors in them, such as incorrect tautomeric forms and protonation states, would affect the final prediction result.
3. Such a machine learning based classification model for recognizing chemicals would be an efficient tool for many other systems in drug discovery and toxicology. Please comment in the manuscript on which possible implementations are likely to occur and clearly outline any potential caveats.
Round 2
Reviewer 2 Report
the author has given justification